# Is sugar as sweet to the palate as seeds are appetizing to the belly? Taste responsiveness to five food-associated carbohydrates in zoo-housed white-faced sakis, *Pithecia pithecia*

**Mikel Redin Hurtado**[1], **Ida Fischer**[2], **Matthias Laska**[1]*

1 Department of Physics, Chemistry and Biology, Linköping University, Linköping, Sweden,
2 Furuviksparken AB, Furuvik, Sweden

* matthias.laska@liu.se

**Data Availability Statement:** The data contained in this manuscript stem from a Master's thesis (of the first author) peformed at Linköping University

## Abstract

Differences in taste perception between species are thought to reflect evolutionary adaptations to dietary specialization. White-faced sakis (*Pithecia pithecia*) are commonly considered as frugivores but are unusual among primates as they do not serve as seed dispersers but rather prey upon the seeds of the fruits they consume and are thought to exploit the lipids and proteins that these seeds contain in high amounts. Using a two-bottle preference test of short duration we therefore assessed whether this dietary specialization affects the taste responsiveness of four adult white-faced sakis for five food-associated carbohydrates. We found that the sakis significantly preferred concentrations as low as 10 mM sucrose, 10–40 mM fructose, 20–30 mM glucose and maltose, and 30–40 mM lactose over tap water. When given the choice between all binary combinations of these five saccharides presented at equimolar concentrations of 100, 200, and 300 mM, respectively, the sakis displayed significant preferences for individual saccharides in the following order: sucrose > fructose > glucose ≥ maltose = lactose. These results demonstrate that seed-predating white-faced sakis have a well-developed taste sensitivity for food-associated carbohydrates which is not inferior to that of most other primates including seed-dispersing frugivores, but rather ranks among the more sweet-taste sensitive species. Further, they show that their pattern of relative preference for the five carbohydrates is similar to that found in other frugivorous primate species. These findings may represent an example of Liem's paradox as the sakis' morphological adaptations to efficiently predate on and exploit the lipid- and protein-rich hard-shelled seeds of fruits does not compromise their ability to detect the carbohydrates found in the pulp of fruits at low concentrations.

## Introduction

Differences in taste perception between species are thought to reflect evolutionary adaptations to dietary specializations [1, 2]. Carnivorous mammals such as cats, for example, have been

(Sweden). The thesis, and thus the complete data, are publicly available at DiVA, a digital repository. (https://liu.diva-portal.org). For accessing the dataset you have to simply type the name of the first author of our manuscript (who is the sole author of the Master's thesis stored in the digital repository) – Mikel Redin Hurtado – and press the "Search" button. In a new window, you can now download a pdf of the thesis. The dataset which forms the basis of our manuscript can be found in Table 1, Fig 2, and Appendix I-III.

**Funding:** The authors received no specific funding for this work.

**Competing interests:** The authors have declared that no competing interests exist.

found to lack a functioning sweet-taste receptor due to a diet-driven relaxed purifying selection pressure on the corresponding gene [3]. Cetaceans such as dolphins even lack all gustatory receptors except those for salty taste and therefore lost the ability to perceive the qualities of sweet, sour, bitter, and umami, presumably because they swallow their fish-based diet whole, without chewing, and thus without gustatory evaluation [4].

In addition to between-species differences in the ability to perceive a given taste quality, comparative studies also reported marked differences in sensitivity for a given taste substance which can be explained by differences in dietary habits. The degree of herbivory, for example, has been shown to correlate negatively with bitter-taste sensitivity in vertebrates [5]. Similarly, nectar-feeding hummingbirds have been reported to be more sensitive to sweet-tasting carbohydrates compared to birds feeding on other diets [6].

Primates appear to be a particularly useful taxon to study between-species differences in taste perception as they include a wide variety of dietary specializations, including frugivores, folivores, herbivores, gummivores, insectivores, graminivores, and omnivores, to name but a few [7]. Accordingly, taste sensitivity to monosodium glutamate, the prototypical umami taste substance, has been reported to correlate negatively with the proportion of animal matter in the diet of nonhuman primates [8]. Similarly, the degree of frugivory has been found to correlate positively with taste sensitivity for food-associated sweet-tasting saccharides among platyrrhine primates [9].

White-faced sakis are platyrrhine primates that live in various types of habitats, including lowland and highland forests, with a clear preference for regions with an abundance of fruit-bearing trees [10]. Their diet has been shown to comprise more than 100 plant species of which 57 were fruit- and thus seed-bearing tree species [11]. They are known to include a large proportion of fruits into their diet and are therefore commonly considered as frugivores [10]. Thus, this platyrrhine primate should be expected to display a high sensitivity for the carbohydrates found in the pulp of fruits they consume. However, in contrast to most other frugivorous primate species, they do not serve as seed dispersers but rather feed on the seeds of the fruits they consume and are thought to exploit the lipids and proteins that these seeds contain in high amounts [12]. Long-term studies on white-faced sakis in Venezuela reported that their diet was composed of 61% seeds, 28% fleshy fruits, 7% leaves, 2% flowers, and 2% of insects, respectively [13]. Other studies corroborated the preponderance of seeds in the diet of white-faced sakis which was found to peak at 94% during the early dry season [Norconk and Conklin-Brittain 2004]. From an energetic point of view this seems reasonable as lipids provide 9 kcal/gram of energy whereas carbohydrates provide only 4 kcal/gram [14]. A recent study supports the notion that white-faced sakis may specialize on exploiting the lipids found in seeds to meet their requirements of metabolic energy as they were shown to display a strong positive correlation between their food preferences and the lipid content of the food types tested whereas all seed-dispersing primate species studied so far display a strong correlation between their food preferences and carbohydrate content [15]. Further, whereas seed-dispersing frugivorous primates usually specialize on ripe fruits, white-faced sakis are known to include unripe fruits into their diet which contain considerably lower amounts of soluble carbohydrates compared to ripe ones [10]. This raises the question whether seed-predating white-faced sakis may be less sensitive to food-associated sweet-tasting carbohydrates compared to seed-dispersing primates.

So far, only very little is known about the sense of taste in white-faced sakis. The only behavioral data reported in this species to date demonstrated that white-faced sakis are unable to perceive the taste of the artificial high-potency sweetener aspartame [16] but prefer solutions of the artificial sweetener alitame over tap water [17]. Cellular assays showed that the T1R1/T1R3 taste receptor of white-faced sakis responds to L-sodium glutamate [18] and that their

Tas2R16 bitter taste receptor responds to the glucosides salicin and helicin [19]. To the best of our knowledge, no behavioral data on the sakis' sensitivity for food-associated taste substances have been published so far.

We therefore decided to determine taste preference thresholds for five food-associated carbohydrates in white-faced sakis. Sucrose, fructose, and glucose typically comprise more than 90% of the total sugar content of the pulp of ripe tropical fruits [20]. Maltose, in contrast, is only rarely found in free form in plant tissues [21] but is released through the action of salivary α-amylase during the oral digestion of starch [22]. Mammals such as rodents and catarrhine primates are therefore likely to perceive the sweetness of maltose when feeding on starch-containing plant parts such as tubers and seeds. Platyrrhine primates, including white-faced sakis, however, do not express salivary α-amylase [23]. Therefore, it is unlikely that maltose may contribute to the sweet sensation perceived by sakis when feeding on seeds. It should thus be interesting to assess whether white-faced sakis are as sensitive to maltose as catarrhine primates are.

Lactose is the main carbohydrate in milk, the first food source of all mammals [24]. Additionally, we assessed relative taste preferences in white-faced sakis for the five carbohydrates mentioned above. Previous studies have found that nonhuman primates also differ in the pattern of relative taste preferences and thus in their perception of the relative sweetness and, concomitantly, of the sweetening potency of naturally occurring soluble carbohydrates, possibly due to differences in their dietary habits [25].

We hypothesized that seed-predating white-faced sakis would be less sensitive to food-associated carbohydrates than seed-dispersing frugivorous primates.

## Methods

### Animals

Four white-faced sakis (*Pithecia pithecia*), maintained at Furuviksparken (Furuvik, Sweden), participated in the study. They comprised two adult males (Kariakov and Engelbrekt, 17 and 11 years of age, respectively) and two adult females (Lisha and Anita, 14 and 7 years of age, respectively). All four animals were born in captivity. The sakis were housed in an indoor enclosure of 633 m$^3$, with access to a 127 m$^3$ outdoor enclosure. The indoor enclosure was connected to a 15 m$^3$ test cage with a sliding door allowing for the temporary separation of the animals for individual testing. Both the indoor and the outdoor enclosure as well as the test cage were equipped with natural vegetation and with numerous structures to climb and perch. Environmental enrichment included structural (e.g. swings, ropes, ladders, boxes), social (the indoor enclosure was designed as a walk-through exhibit for visitors, and shared with iguanas and turtles), and food (e.g. food items were hidden to elicit foraging behavior) enrichment.

The animals were fed fresh fruits and vegetables, seeds, nuts, insects, and a tamarin cake (from Mazuri Zoo Foods, Witham, Essex, Great Britain). Additionally, edible fresh leaves, branches and flowers were presented in the enclosure *interdum*. Fruits and vegetables were fed twice each day around 08:00 and 15:00, respectively, while water was always available *ad libitum*. Food remnants were still present the following morning, suggesting that it was unlikely that voracious appetite affected the outcome of the taste tests performed in this study.

### Ethical note

The experiments reported here comply with the *American Society of Primatologists' Principles for the Ethical Treatment of Primates*, with the *European Union Directive on the Protection of Animals Used for Scientific Purposes* (EU Directive 2010/63/EU), and with current Swedish

animal welfare laws. The ethical board of Furuviksparken AB approved the study prior to its commencement (Protocol number: 2022–4).

## Taste substances

We used the following five carbohydrates: sucrose (CAS# 57-50-1), fructose (CAS# 57-48-7), glucose (CAS# 50-99-7), maltose (CAS# 6363-53-7) and lactose (CAS# 63-42-3). All substances were obtained from Sigma-Aldrich (St. Louis, MO) and were of the highest available purity ($\geq$99.5%).

## Procedures

We employed a two-bottle preference test of short duration [26]. The animals were allowed for 2 minutes to drink from a pair of simultaneously presented 150 ml cylinders with metal drinking spouts. We performed up to six trials per day and animal, starting approximately 2 hours after the morning feeding, respectively. Care was taken to keep intertrial intervals of at least 45 minutes between consecutive trials. Tests were conducted on 5–6 days per week.

## Determination of taste preference thresholds

Here, the animals were given the choice between tap water and defined concentrations of a given carbohydrate dissolved in tap water. With all substances, testing started at a concentration of 200 mM and proceeded through 100, 50, 20, 10 mM, etc. until an individual failed to show a significant preference for the taste solution over the tap water. Subsequently, solutions with intermediate concentrations (between the lowest concentration that was preferred and the first concentration that was not) were used in order to determine preference threshold values more exactly. Testing did not follow a strict descending staircase procedure but a pseudo-randomized order between presumably more and less attractive concentrations to maintain the animals' willingness to cooperate. The order in which the five carbohydrates were tested was the same for all four animals (1. sucrose, 2. fructose, 3. glucose, 4. maltose, 5. lactose).

## Assessment of relative sweetness

Here, the animals were given the choice between two carbohydrate solutions presented at equimolar concentrations. All ten possible binary stimulus combinations (e.g. sucrose *versus* glucose, fructose *versus* maltose, etc.) were tested. Three test series were performed at 100, 200, and 300 mM, respectively, to assess whether preferences were stable at different concentration levels.

In both experiments, each pair of stimuli was presented 10 times per individual, with the position of the bottles pseudo-randomized to counteract possible position biases. During each trial, care was taken to ensure that an animal sampled both bottles at least once. The animals were trained to drink voluntarily from the pair of simultaneously presented bottles and were completely accustomed to the procedure.

## Data analysis

For each animal, the amount of liquid consumed from each bottle was recorded, summed for the 10 test trials with a given stimulus combination, converted to percentages (relative to the total amount of liquid consumed from both bottles), and 66.7% (i.e., 2/3 of the total amount of liquid consumed) was taken as criterion of preference. This rather conservative criterion was adopted for purposes of comparability with previous studies using the same criterion in other primate species [9, 25, 27–31] and in order to avoid misinterpretation of data due to a too liberal criterion.

In addition, binomial tests were conducted, and an animal was only regarded as significantly preferring one of the two stimuli if it reached the criterion of 66.7% and consumed more from the bottle containing the taste solution in $\geq 8$ of 10 trials (binomial test, p < 0.05). Taste preference thresholds were defined as the lowest concentration at which the animal met both criteria mentioned above.

## Results

### Taste preference thresholds

Taste preference thresholds of the four white-faced sakis were found to be 10 mM for sucrose, 10–40 mM for fructose, 20–30 mM for glucose and maltose, and 30–40 mM for lactose (Fig 1). All animals failed to show a significant preference for the lowest concentrations presented, suggesting that the preference for higher concentrations was indeed based on the chemical nature of the stimuli. Interindividual variability of taste preference threshold values with a given substance was low and ranged between zero (i.e., all four animals displaying the same threshold value) with sucrose and a dilution factor of 4 between the most- and the least-responsive animal with fructose.

### Relative sweetness

When given the choice between two aqueous carbohydrate solutions presented at equimolar concentrations of 100, 200, and 300 mM, respectively, the white-faced sakis as a group significantly preferred sucrose over all other carbohydrates, and fructose over glucose, maltose, and lactose (Fig 2). This was also true when the four animals were considered separately. At 300 mM, i.e. the highest concentration tested, the sakis also showed a clear preference for glucose over maltose. Here, too, interindividual variability was low and with only few exceptions all four animals either reached the criterion of preference ($\geq 66.7\%$ of total consumption, plus binomial test, p < 0.05) with a given stimulus combination or all four animals failed to do so.

### Behavioral observations

Although we did not systematically record or analyse the frequency of alternations between bottles in a given 2-min trial, the sakis showed a trend for more alternations between bottles in trials when lower concentrations of the taste stimuli were presented compared to trials in which higher concentrations were presented. With higher and thus more attractive concentrations of the saccharide solutions the sakis always drank for the full 2 minutes per trial whereas they occasionally stopped drinking from lower and thus less attractive concentrations prior to the end of a trial. Similarly, when presented with higher concentrations of a saccharide solution the animals were always eager to participate in the subsequent trial whereas they were occasionally reluctant to participate if the previous trial included lower concentrations. There was no indication that the animals' food intake behavior at regular feeding times would be affected by the introduction of regular taste preference tests. We also failed to notice any difference in food intake behavior of the sakis between days with and days without performing the taste tests.

## Discussion

The results of the present study demonstrate that white-faced sakis have a well-developed taste sensitivity for food-associated carbohydrates which is not inferior to that of most other primates, but rather ranks among the more sweet-taste sensitive species. Further, they show that the pattern of relative preference for the five carbohydrates displayed by the sakis is similar to that found in other frugivorous primate species.

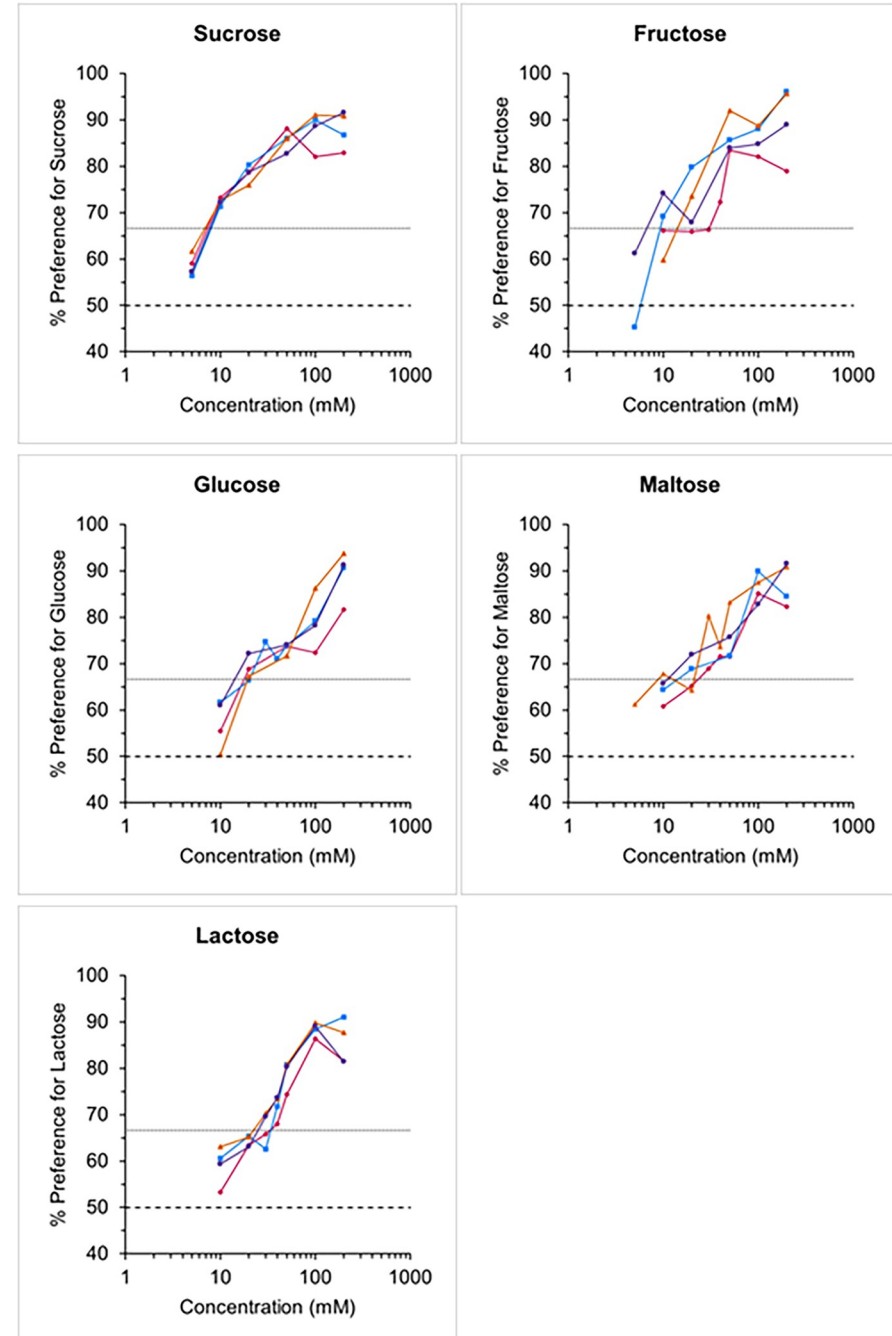

**Fig 1. Taste responses of four white-faced sakis to aqueous solutions of sucrose, fructose, glucose, maltose, and lactose tested against tap water.** Each data point represents the mean value of 10 trials of 2 min per animal. Each symbol represents one of the four individuals (circle: Engelbrekt; square: Lisha; triangle: Anita; diamond: Kariakov). The dashed horizontal lines at 66.7% and at 50% indicate the criterion of preference and the chance level, respectively.

## Taste preference thresholds

The taste preference threshold values of the white-faced sakis for the five food-associated carbohydrates fall generally into the lower range compared to those reported in other primates, suggesting a comparatively high sweet-taste sensitivity (Table 1). This is in contrast to our

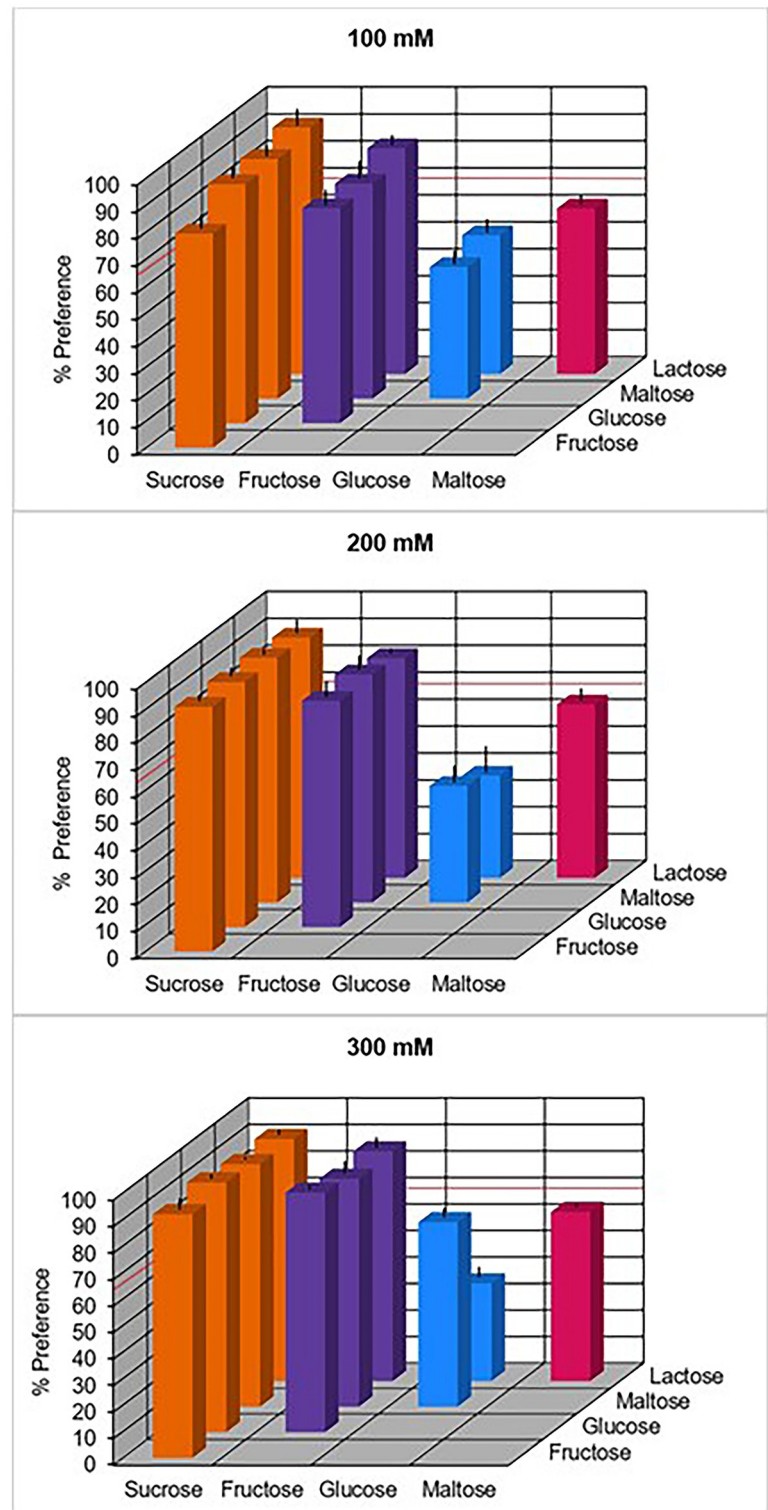

**Fig 2. Relative taste preferences of four white-faced sakis when given the choice between two aqueous carbohydrate solutions presented at equimolar concentrations of 100, 200, and 300 mM, respectively.** Each bar represents the mean preference (±SD) from 10 trials of 2 minutes per individual for the carbohydrate on the left relative to the carbohydrate on the right. The red line at 66.7% indicates the criterion of preference.

**Table 1. Taste preference thresholds for food-associated carbohydrates in primates (mM).**

| Species | Sucrose | Fructose | Glucose | Maltose | Lactose | Ref. |
|---|---|---|---|---|---|---|
| **Strepsirrhine primates** | | | | | | |
| *Varecia variegata* | 25 | 25 | 75 | 50 | 50 | [28] |
| *Eulemur coronatus* | | 21 | | | | [37] |
| *Eulemur fulvus* | 9 | 22.5 | | | | [37] |
| *Eulemur macaco* | 8 | 14 | | | | [37] |
| *Eulemur mongoz* | 125 | 110 | | | | [38] |
| *Hapalemur simus* | 17.5 | 18.5 | | | | [37] |
| *Hapalemur griseus* | | 16.5 | | | | [37] |
| *Phaner furcifer* | 65 | | | | | [37] |
| *Microcebus murinus* | 167 | 47.5 | | | | [38, 39] |
| *Microcebus coquereli* | 90 | | | | | [37] |
| *Cheirogaleus major* | 50 | | | | | [37] |
| *Cheirogaleus medius* | 143 | 100 | | | | [38, 40] |
| *Propithecus verreauxi* | 52.5 | | | | | [37] |
| *Loris tardigradus* | 50 | | | | | [38] |
| *Nycticebus coucang* | 330 | | | | | [38] |
| *Galago senegalensis* | 66 | | | | | [38] |
| **Platyrrhine primates** | | | | | | |
| *Pithecia pithecia* | **10** | **10–40** | **20–30** | **20–30** | **30–40** | |
| *Ateles geoffroyi* | 3 | 15 | 20 | 20 | 10 | [9] |
| *Saimiri sciureus* | 10 | 40 | 90 | 90 | 100 | [27] |
| *Saguinus midas* | 66 | 66 | 330 | | 250 | [38] |
| *Saguinus fuscicollis* | 50 | | | | | [38] |
| *Saguinus oedipus* | 125 | 16 | | | | [37, 38] |
| *Cebuella pygmaea* | 33 | 50 | 100 | | 125 | [38] |
| *Callithrix jacchus* | 25 | 29.5 | | | | [38, 41] |
| *Callithrix geoffroyi* | | 41 | | | | [37] |
| *Callithrix argentata* | | 19.5 | | | | [37] |
| *Leontopithecus rosalia* | | 19.5 | | | | [37] |
| *L. chrysomelas* | | 21.5 | | | | [37] |
| *Callimico goeldii* | | 31 | | | | [37] |
| *Cebus apella* | 8 | | | | | [37] |
| *Aotus trivirgatus* | 17 | | | | | [38] |
| **Catarrhine primates** | | | | | | |
| *Macaca nemestrina* | 10 | 20 | 20 | 10 | 30 | [42] |
| *Macaca mulatta* | 6 | | | | | [38] |
| *Macaca radiata* | 10 | | | 10 | | [43] |
| *Macaca fuscata* | 10 | | | 10 | | [44] |
| *Papio hamadryas* | 10 | 20 | 25 | 20 | 20 | [45] |
| *Cercopithecus pygerythrus* | 11 | | | | | [38] |
| *Cercopithecus nictitans* | 11 | | | | | [38] |
| **Hominoid primates** | | | | | | |
| *Pongo pygmaeus* | | 15 | | | | [46] |
| *Pan troglodytes* | 20 | 40 | 80 | 80 | 80 | [25] |
| *Gorilla gorilla* | | 75 | | | | [46] |

(*Continued*)

**Table 1.** (Continued)

| Species | Sucrose | Fructose | Glucose | Maltose | Lactose | Ref. |
|---|---|---|---|---|---|---|
| *Homo sapiens* | 10 | 40 | 80 | 31 | 72 | [35] |

Data of the present study are given in **bold** typeface. Please note that the values presented for *Homo sapiens* are taste *detection* thresholds, and not taste *preference* thresholds.

hypothesis that seed-predating white-faced sakis would be less sensitive to food-associated carbohydrates than seed-dispersing frugivorous primates.

Chimpanzees (*Pan troglodytes*), for example, are considered as seed-dispersing ripe-fruit specialists [32] but nevertheless display higher taste preference threshold values, and thus a lower sensitivity, with all five saccharides than the white-faced sakis of the present study [25]. Similarly, black-and-white ruffed lemurs (*Varecia variegata*) are less sensitive for all five carbohydrates compared to *Pithecia pithecia* [28], despite being highly frugivorous seed-dispersers [33]. Only the black-handed spider monkey (*Ateles geoffroyi*), a seed-dispersing and highly frugivorous platyrrhine primate species [34], tends to have lower threshold values, and thus a higher sweet-taste sensitivity, than the white-faced sakis [9].

Our finding that the taste preference thresholds of the white-faced sakis are identical with (sucrose) of even lower than (glucose, maltose, and lactose) the taste detection threshold values reported in human subjects [35] is remarkable considering that the sophisticated signal detection methods used in taste tests with human subjects generally lead to lower threshold values compared to the simple two-bottle preference tests employed with nonhuman primates which provide only a conservative approximation of a species' taste sensitivity [36]. Accordingly, we conclude that white-faced sakis are at least as sensitive as, and possibly even more sensitive than, human subjects for the food-associated carbohydrates employed here.

The taste preference thresholds of the white-faced sakis for sucrose (10 mM), fructose (10–40 mM), and glucose (20–30 mM) are markedly lower than the concentrations of these saccharides found in ripe tropical fruits that primates are known to feed on in the wild [47]. Accordingly, these saccharides should be clearly detectable for the sakis when feeding on fruits, supporting the notion that the sweetness of fruits may serve as a criterion for consumption in primates [48].

The lactose content of the milk of white-faced sakis is, to the best of our knowledge, yet unknown. However, it is reasonable to assume that the lactose content of saki milk should not be fundamentally different from that of other platyrrhine primates which ranges between 6–9 g/100 ml, corresponding to 175–263 mM [49]. These lactose concentrations are substantially higher than the taste preference threshold values of 30–40 mM for lactose determined in the present study. This, in turn, suggests that infant white-faced sakis should be clearly able to perceive the sweetness of lactose when feeding on breast milk.

Maltose is only rarely found in free form in fruits or other parts of plants [21] but is released through the action of salivary α-amylase during the mastication of plant parts containing starch [22]. Mammals such as rodents and catarrhine primates are therefore likely to perceive the sweetness of maltose when feeding on starch-containing plant parts such as tubers and seeds. However, platyrrhine primates, including white-faced sakis, do not express salivary α-amylase [23]. Therefore, it is unlikely that maltose may contribute to the sweet sensation perceived by sakis when feeding on seeds. It is thus surprising that the white-faced sakis display a taste preference threshold value for maltose (20–30 mM) which is almost as low as that of catarrhine primates that are known to include a high proportion of starch-rich plant material into their diet such as members of the genus *Macaca* (10 mM) [42, 44] and the hamadryas

**Table 2. Relative taste preferences for food-associated carbohydrates in primates and non-primate mammals.**

| Species | Relative taste preference | Ref. |
| --- | --- | --- |
| **Strepsirrhine primates** | | |
| *Varecia variegata* | sucrose > fructose > glucose ≥ maltose ≥ lactose | [28] |
| **Platyrrhine primates** | | |
| ***Pithecia pithecia*** | **sucrose > fructose > glucose ≥ maltose = lactose** | |
| *Ateles geoffroyi* | sucrose > fructose > glucose ≥ lactose ≥ maltose | [56] |
| *Saimiri sciureus* | sucrose > fructose > glucose ≥ maltose ≥ lactose | [55] |
| **Catarrhine primates** | | |
| *Macaca nemestrina* | maltose > sucrose > glucose ≥ fructose ≥ lactose | [42] |
| **Hominoid primates** | | |
| *Pan troglodytes* | sucrose > fructose > glucose = maltose = lactose | [25] |
| *Homo sapiens* | sucrose > fructose > maltose ≥ glucose ≥ lactose | [57] |
| **Non-primate mammals** | | |
| *Oryctolagus cuniculus* | maltose > sucrose > glucose > fructose ≥ lactose | [59] |
| *Rattus norvegicus* | maltose > sucrose = glucose > lactose | [26] |
| *Rattus norvegicus* | maltose > sucrose > glucose = fructose | [60] |

Data of the present study are given in **bold** typeface.

baboon (*Papio hamadryas*) (20 mM) [45]. However, white-faced sakis, like all mammals, express pancreatic α-amylase and are therefore able to hydrolyse starch into maltose in their intestines. Accordingly, they can exploit the metabolic energy from the starch which is found in considerable amounts in the seeds consumed by sakis [50].

## Relative sweetness

The food-associated carbohydrates used in the present study are known to differ in their stimulating efficiency when interacting with a mammalian sweet-taste receptor [51]. Accordingly, they differ in their sweetening potency, that is, in their subjectively perceived sweetness when presented at equimolar concentrations [52]. Differences in the attractiveness of sweet-tasting substances as displayed in two-bottle preference tests are thought to reflect differences in their stimulating efficiency and, concomitantly, in their perceived sweetness. Both electrophysiological findings [53, 54] and behavioral studies [55, 56] in nonhuman primates support this notion. The relative taste preferences found in the present study can therefore be considered as an approximation of the relative sweetness of the five saccharides as perceived by the white-faced sakis. The pattern of relative taste preference displayed by the sakis (sucrose > fructose > glucose ≥ maltose = lactose) is identical or at least very similar to that found in previous studies with spider monkeys [56], squirrel monkeys [55], black-and-white ruffed lemurs [28], and chimpanzees [25] (Table 2). It is also similar to the pattern of relative sweetness as perceived by human subjects [57]. All these species clearly prefer sucrose over the four other saccharides tested, and they all prefer fructose over glucose, maltose, and lactose when presented at equimolar concentrations. It is interesting to note that pigtail macaques (*Macaca nemestrina*) clearly differ in this respect as, similar to rats and rabbits, but unlike the primate species mentioned above, they prefer maltose over sucrose [42]. The diet of rats, rabbits, and pigtail macaques is known to contain a high proportion of starch, either in the form of seeds [50] or in the form of starch-containing underground storage organs such as roots, tubers, and bulbs [58]. Thus, the present findings support the notion that relative preferences for food-associated carbohydrates may correlate with dietary specialization in primates.

## Taste responsiveness and seed predation

Seed predation in primates is considered as a derived trait that evolved from frugivory as the ancestral condition [61]. White-faced sakis show marked morphological adaptations to seed predation, indicating the importance of seeds as a source of metabolic energy and, possibly, other critical nutrients. The teeth of sakis, for example, include procumbent incisors and robust, laterally splayed canines which facilitate the extraction of seeds protected by tough outer membranes [62]. Their molars are low-crowned and their crenulated enamel is thought to facilitate stabilization of seeds during crushing [63]. Further, the mandibles of sakis are markedly more robust compared to those of seed-dispersing frugivorous primates which is a prerequisite for the attachment of the massive muscles that are needed to provide the necessary pressure to crack open hard-shelled seeds [64].

Despite these apparent morphological adaptations to the predation of seeds which contain hardly any soluble saccharides [65], the white-faced sakis of the present study displayed a sweet-taste sensitivity for food-associated carbohydrates that is not inferior to but rather at least as high as that of most frugivorous seed-dispersing primates tested so far. This might represent an example of Liem's paradox as the sakis' morphological adaptations suggest that they may exploit the lipids and proteins of the seeds they consume for meeting their requirements of metabolic energy rather than the carbohydrates of the fruit pulp. Liem's paradox refers to the observation that species with seemingly specialized phenotypes can sometimes behave as ecological generalists [66]. For instance, numerous cichlid fish species with elaborate morphological adaptations for feeding specializations were found to frequently feed on prey items for which their specializations are seemingly not adapted [67]. In other words: a highly derived morphology does not necessarily indicate a specialized diet.

Optimal foraging theory may provide a plausible explanation for this seeming mismatch between morphological adaptation and diet: Robinson and Wilson [68] proposed that some food resources such as the pulp of ripe fruit are easy to exploit and are therefore widely preferred among primates while other food resources such as hard-shelled seeds and unripe fruits require specialized phenotypic traits (e.g. modified dentition and masticatory muscles) on the part of the consumer. This allows optimally foraging consumers to evolve phenotypic specializations on generally nonpreferred resources such as seeds without compromising their ability to also exploit preferred resources. The most important advantage of seed predation in primates is thought to be related to a minimized fluctuation in plant resource availability and a broadened niche relative to seed-dispersing competitors in gaining access to the nutrients of seeds [69]. This seems plausible considering that sakis are known to include unripe fruit into their diet even during seasons when ripe fruits are available–presumably to avoid food competition with sympatric ripe-fruit specialists [13]—and that young seeds in unripe fruits already contain high amounts of lipids and proteins [65] whereas the pulp of unripe fruits usually contains markedly lower amounts of carbohydrates than ripe ones [10].

In conclusion, the results of the present study suggest that the dietary specialization of white-faced sakis as seed predators does not negatively affect their ability to detect food-associated carbohydrates found in the pulp of fruits at concentrations that are as low as those reported in seed-dispersing frugivorous primates.

## Acknowledgments

The authors gratefully acknowledge the help of the primate caretakers at Furuviksparken.

## Author Contributions

**Conceptualization:** Mikel Redin Hurtado, Matthias Laska.

**Formal analysis:** Mikel Redin Hurtado, Matthias Laska.

**Investigation:** Mikel Redin Hurtado, Ida Fischer.

**Methodology:** Mikel Redin Hurtado, Ida Fischer, Matthias Laska.

**Resources:** Ida Fischer, Matthias Laska.

**Visualization:** Mikel Redin Hurtado, Ida Fischer, Matthias Laska.

**Writing – original draft:** Mikel Redin Hurtado, Ida Fischer, Matthias Laska.

**Writing – review & editing:** Mikel Redin Hurtado, Ida Fischer, Matthias Laska.

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
