## [Decision Letter · Decision Letter 0]

29 Aug 2023

PONE-D-23-22500Is sugar as sweet to the palate as seeds are appetizing to the belly? Taste responsiveness to five food-associated carbohydrates in zoo-housed white-faced sakis, Pithecia pitheciaPLOS ONE

Dear Dr. Laska,

Thank you for submitting your manuscript to PLOS ONE. After careful consideration, we feel that it has merit but does not fully meet PLOS ONE’s publication criteria as it currently stands. Therefore, we invite you to submit a revised version of the manuscript that addresses the points raised during the review process.

We look forward to receiving your revised manuscript.

Kind regards,

Nicoletta Righini, PhD

Academic Editor

PLOS ONE

Journal Requirements:

https://link.springer.com/article/10.1007/s10329-018-0697-0

In your revision ensure you cite all your sources (including your own works), and quote or rephrase any duplicated text outside the methods section. Further consideration is dependent on these concerns being addressed.

**Additional Editor Comments (if provided):**

In this manuscript the authors carried out preference tests to assess taste responsiveness to sugars in captive white-faces sakis. The ms is well written and the aim of the study is straightforward and well justified. The reviewers suggest further elaboration of some topics, including improved justifications, more data/examples on feeding behavior and diets of wild sakis, and more details on the behavior of the study subjects.

Reviewers' comments:

Reviewer's Responses to Questions

**Comments to the Author**

1. Is the manuscript technically sound, and do the data support the conclusions?

Reviewer #1: Partly

Reviewer #2: Yes

2. Has the statistical analysis been performed appropriately and rigorously? 

Reviewer #1: Yes

Reviewer #2: I Don't Know

3. Have the authors made all data underlying the findings in their manuscript fully available?

Reviewer #1: Yes

Reviewer #2: Yes

4. Is the manuscript presented in an intelligible fashion and written in standard English?

Reviewer #1: Yes

Reviewer #2: Yes

5. Review Comments to the Author

**Reviewer #1: **Authors argue that sakis may have a lower taste sensibility to carbohydrates than most frugivorous non-human primates (because of sakis also may use seeds of the fruits they consume as food and obtain energy), even they show that “white-faced sakis include a large proportion of fruits into their diet and are therefore commonly considered as frugivores”.

The information described in the introduction section does not fully justify this idea, as authors don’t show data of saki’s feeding behavior as feeding rate, specifically on ripe/unripe fruits/seeds and compare this data with others frugivorous seed-dispersing monkeys. Why sakis should be less sensitive to carbohydrates if simple sugars are the main source of easily digestible energy? I invite the authors to reinforce their hypothesis.

Authors determined taste preference thresholds for five food-associated carbohydrates, although, I wonder if sakis secrete salivary amylase, as they used maltose and they didn’t show if this sugar can be found in diet of sakis, or if sakis may secrete salivary amylase or maltase which help to release simple sugars.

Annotations:

L74. Please explain with specific examples, feeding time (%)- units/min sakis include unripe fruits into their diet and compare with others frugivorous primates. What if sakis include unripe fruits into their diet due low availability of ripe fruits? Which level of no ripeness do sakis accept?

L91. Do sakis secrete salivary amylase? Which arboreal plant species contain starch? Please give examples, and justify using Maltose as one of your test carbs.

L113. Please show nutritional content of the diet and food consumption to ensure that caloric requirements were met, so you can suggest it was unlikely that voracious appetite affected the outcome of the taste tests.

**Reviewer #2: **This is a well-written and informative manuscript reporting the results of a study of taste preference in white-faced sakis. This species is of interest in the context of sweet taste sensitivity in light of their preferential consumption of fruit seeds rather than fruit flesh (and thus greater lipid > carbohydrate intake). The choice of sugars to evaluate is sensible (sucrose/fructose/glucose/maltose/lactose) and results indicate, contrary to hypotheses, an ordered set of preferences that aligns with other monkey species that consume fruit flesh.

The results are presented clearly and interpreted with insightful comments. I therefore have only minor comments and a few queries.

1. It would be illuminating to briefly describe the habitat and sorts of fruits that comprise the diet of the white-faced saki in the introduction.

2. It is stated that trials began ~2h after feeding - but was this after the 0800 or 1500 feed (or both)? In addition, please specify on how many days per week tests were conducted.

3. Did the researchers track the pattern of alternations between the two solutions? This would be interesting to know, even if just informal observations. Rodent data show that animals will periodically switch between two tastants under some conditions.

4. Were there any other noteworthy behavioural responses to the tastants such as frustration with diluted concentrations or enhanced appetitive/approach responses after initial consumption?

5. Notwithstanding the small sample, could the authors comment on difference between the sexes or according to age?

6. Did the introduction of regular taste preference tests alter the subjects' food intake behaviour at regular feeding sessions?

7. Is there a possibility that the diet of wild white-faced sakis might differ from the subjects of the present study, and if so, might this be a caveat worth mentioning?

8. In a similar vein to the discussion of optimal foraging theory, is there any relationship between sweetness of unripe fruit flesh and the lipid content (i.e. energy density) of the seed?

9. Greater detail about the statistical analyses would be useful for clarity.

Minor comments

Line 66: is 'prey upon' correct terminology here? 'Consume' or 'feed on' seems more appropriate as seeds are inanimate.

Line 92: 'Additionally' does not seem to flow here. Suggest replacing with 'Thus'

6. PLOS authors have the option to publish the peer review history of their article (what does this mean?). If published, this will include your full peer review and any attached files.

Reviewer #1: No

Reviewer #2: No

---

## [Author Response · Author response to Decision Letter 0]

12 Sep 2023

Response to the reviewers’ comments

Reviewer #1: Authors argue that sakis may have a lower taste sensibility to carbohydrates than most frugivorous non‐human primates (because of sakis also may use seeds of the fruits they consume as food and obtain energy), even they show that “white‐faced sakis include a large proportion of fruits into their diet and are therefore commonly considered as frugivores”. The information described in the introduction section does not fully justify this idea, as authors don’t show data of saki’s feeding behavior as feeding rate, specifically on ripe/unripe fruits/seeds and compare this data with others frugivorous seed‐dispersing monkeys. Why sakis should be less sensitive to carbohydrates if simple sugars are the main source of easily digestible energy? I invite the authors to reinforce their hypothesis.

Response: We took up the reviewer’s suggestion and now provide data on the proportion of seeds, fruits, and other food types consumed by white-faced sakis in the wild. Although the literature does not provide exact percentages of ripe versus unripe fruit consumption in white-faced sakis, there is broad consensus that seed-predating sakis differ from seed-dispersing platyrrhine primates - which are usually ripe-fruit specialists - in including unripe fruits into their diet. We mention this explicitly in the Introduction section. 

Authors determined taste preference thresholds for five food‐associated carbohydrates, although, I wonder if sakis secrete salivary amylase, as they used maltose and they didn’t show if this sugar can be found in diet of sakis, or if sakis may secrete salivary amylase or maltase which help to release simple sugars.

Response: We took up the reviewer’s suggestion and now mention not only in the Discussion section, but also in the Introduction that platyrrhine primates, including white-faced sakis, do not express salivary �-amylase and that it is therefore unlikely that maltose may contribute to the sweetness perceived by sakis when feeding on starch-rich seeds. Further, we now mention in the Discussion section that sakis, like all mammals, express pancreatic �-amylase and are therefore able to hydrolyse starch into maltose in their intestines. Further, we now mention more clearly that starch is found in considerable amounts in the seeds consumed by sakis and that they are therefore able to exploit the metabolic energy from this starch. 

Annotations:

L74. Please explain with specific examples, feeding time (%)‐ units/min sakis include unripe fruits into their diet and compare with others frugivorous primates. What if sakis include unripe fruits into their diet due low availability of ripe fruits? Which level of no ripeness do sakis accept?

Response: As mentioned further above, the literature does not provide exact percentages of ripe versus unripe fruit consumption in white-faced sakis. However, there is broad consensus that seed-predating sakis differ from seed-dispersing platyrrhine primates - which are usually ripe-fruit specialists - in including unripe fruits into their diet. We mention this explicitly in the Introduction section. Further, we mention in the Discussion section that sakis evolved modified dentition and masticatory muscles allowing them to feed on hard-shelled seeds. We now broadened this statement and mention that these morphological adaptations are also advantageous to feed on unripe fruits which are usually harder compared to ripe fruits. We now mention in the Discussion section that white-faced sakis do not only feed on unripe fruits during times of low availability of ripe fruits, but rather also do so during seasons when ripe fruits are available – presumably to avoid food competition with sympatric ripe-fruit specialists. To the best of our knowledge, no study so far assessed “which level of no ripeness sakis accept”.

L91. Do sakis secrete salivary amylase? Which arboreal plant species contain starch? Please give examples, and justify using Maltose as one of your test carbs.

Response: We rephrased the corresponding statement in the Introduction section and now mention that platyrrhine primates, including white-faced sakis, do not express salivary �-amylase and that it is therefore unlikely that maltose may contribute to the sweetness perceived by sakis when feeding on seeds. We now mention more clearly that starch is found in considerable amounts in the seeds consumed by sakis. Further, we now mention that the lack of salivary �-amylase should make it interesting to assess whether the sakis’ taste sensitivity for maltose differs from that of primates that do express salivary �-amylase.

L113. Please show nutritional content of the diet and food consumption to ensure that caloric requirements were met, so you can suggest it was unlikely that voracious appetite affected the outcome of the taste tests.

Response: We carefully considered the reviewer’s suggestion. However, we do not feel that it would be necessary or helpful for the reader to show the nutritional content of the sakis’ diet. The fact that the animals were in good health, did not display any morphological, physiological or behavioral signs of malnutrition and had an appropriate body weight (all regularly monitored by a veterinarian) plus the mentioned fact that food remnants were still present the following morning should be sufficient to support our statement that it was unlikely that voracious appetite affected the outcome of the taste tests.

Reviewer #2: This is a well‐written and informative manuscript reporting the results of a study of taste preference in white‐faced sakis. This species is of interest in the context of sweet taste sensitivity in light of their preferential consumption of fruit seeds rather than fruit flesh (and thus greater lipid > carbohydrate intake). The choice of sugars to evaluate is sensible (sucrose/fructose /glucose/maltose/lactose) and results indicate, contrary to hypotheses, an ordered set of preferences that aligns with other monkey species that consume fruit flesh. The results are presented clearly and interpreted with insightful comments. I therefore have only minor comments and a few queries.

1. It would be illuminating to briefly describe the habitat and sorts of fruits that comprise the diet of the white‐faced saki in the introduction.

Response: We took up the reviewer’s suggestion and now added some information on the habitat and types of plants that the sakis feed on in the wild to the Introduction section.

2. It is stated that trials began ~2h after feeding ‐ but was this after the 0800 or 1500 feed (or both)? In addition, please specify on how many days per week tests were conducted.

Response: We took up the reviewer’s suggestion and now added information that the trials began approximately two hours after the 08:00 morning feeding. Further, we now added information that tests were conducted on 5-6 days per week.

3. Did the researchers track the pattern of alternations between the two solutions? This would be interesting to know, even if just informal observations. Rodent data show that animals will periodically switch between two tastants under some conditions.

Response: We are grateful for the reviewer’s questions concerning the sakis’ behavior during and after the two-bottle taste tests (points 3,4, and 6) and decided to add a short paragraph entitled “Behavioral observations” to the Results section.

As mentioned in the Methods section, we took care to ensure that an animal sampled both bottles at least once during each trial. Although we did not systematically record or analyse the frequency of alternations between bottles in a given trial, there was a trend for more alternations between bottles when lower concentrations of the taste stimuli were presented compared to higher concentrations. 

4. Were there any other noteworthy behavioural responses to the tastants such as frustration with diluted concentrations or enhanced appetitive/approach responses after initial consumption?

Response: We took up the reviewer’s suggestion and now added information on this in the new Results paragraph “Behavioral observations”. With higher and thus attractive concentrations of the saccharide solutions the sakis always drank for the full 2 minutes per trial whereas they occasionally stopped drinking from lower and thus less attractive concentrations prior to the end of a 2-min trial. Similarly, with higher concentrations the animals were always eager to participate in the subsequent trial whereas they were occasionally reluctant to participate if the previous trial included only lower concentrations.

5. Notwithstanding the small sample, could the authors comment on difference between the sexes or according to age?

Response: We informally analysed whether our results might suggest systematic sex or age differences in sweet-taste sensitivity or in perception of relative sweetness but failed to find any. Considering our small sample size, we decided not to mention this in the text. 

6. Did the introduction of regular taste preference tests alter the subjects' food intake behaviour at regular feeding sessions?

Response: We took up the reviewer’s suggestion and now added information on this in the new Results paragraph “Behavioral observations”. The animals’ food intake behavior at regular feeding times was not at all affected by the introduction of regular taste preference tests. 

7. Is there a possibility that the diet of wild white‐faced sakis might differ from the subjects of the present study, and if so, might this be a caveat worth mentioning?

Response: Yes, the diet of white-faced sakis in the wild does of course differ from the diet fed to the captive animals of the present study, at least with regard to the plant species consumed/offered. However, the food offered to our captive sakis strictly followed the EAZA recommendations for the captive diet of this species and meets the nutritional requirements of sakis as suggested in the literature (e.g. National Research Council, Nutrient requirement of nonhuman primates, 2nd ed., National Academies Press, Washington DC, 2003). Accordingly, the nutrient composition of the captive diet fed to our sakis was as similar to the nutrient composition that sakis feed on in the wild as possible. 

We would not be aware of studies suggesting that the diet fed to experimental animals would affect their sweet-taste sensitivity and their perception of relative sweetness – unless a grossly unbalanced diet is fed for a long time so that animals suffer from the effects of malnutrition. As this was not the case in the present study, we do not feel that it would be worthwhile discussing this point.

8. In a similar vein to the discussion of optimal foraging theory, is there any relationship between sweetness of unripe fruit flesh and the lipid content (i.e. energy density) of the seed?

Response: We already mention in the Discussion section that young seeds in unripe fruits already contain high amounts of lipids and proteins whereas the pulp of unripe fruits usually contains markedly lower amounts of carbohydrates than ripe ones.

9. Greater detail about the statistical analyses would be useful for clarity.

Response: We carefully considered the reviewer’s suggestion. However, we feel that we described our analysis of data in sufficient detail. Please note that we employed an, admittedly, arbitrary but widely used criterion of preference (66.7%, that is, 2/3 of the total amount of liquid consumed) and that we added a second, statistically-based criterion by applying the binomial test (consuming more from a given bottle in ≥ 8 of 10 trials). Due to the limited number of animals in our, and in most other studies on taste responsiveness in primates, it is not possible to apply further statistical tests.

Minor comments

Line 66: is 'prey upon' correct terminology here? 'Consume' or 'feed on' seems more appropriate as seeds are inanimate.

Response: We took up the reviewer’s suggestion and replaced “prey upon” with “feed on” – which may be easier to understand for non-expert readers. However, we would like to point out that the expressions “seed predation” and “preying upon seeds” are commonly used in the context of animals exploiting the nutrients provided by seeds – as opposed to “seed dispersal” which describes the passage of seeds through an animal’s digestive tract without exploiting their nutrients.

Line 92: 'Additionally' does not seem to flow here. Suggest replacing with 'Thus'

Response: We politely disagree on this minor point. In the previous sentences of the paragraph in question we explain to the reader that we determined taste preference thresholds (i.e. peformed sugar versus water tests) and we mention the five carbohydrates employed. This could be considered as “Experiment 1” of the present study. Then we mention that we additionally assessed relative taste preferences (i.e. performed sugar A versus sugar B tests) for the same five carbohydrates. This could be considered as “Experiment 2” of the present study. Accordingly, we feel that “Additionally” fits in this context.

---

## [Editor Report · Decision Letter 1]

14 Sep 2023

Is sugar as sweet to the palate as seeds are appetizing to the belly? Taste responsiveness to five food-associated carbohydrates in zoo-housed white-faced sakis, Pithecia pithecia

PONE-D-23-22500R1

Dear Dr. Laska,

We’re pleased to inform you that your manuscript has been judged scientifically suitable for publication and will be formally accepted for publication once it meets all outstanding technical requirements.

Kind regards,

Nicoletta Righini, PhD

Academic Editor

PLOS ONE

Additional Editor Comments (optional):

The authors clarified and responded adequately to all the queries of the reviewers. The study, even if with a small sample size, adds valuable information to the field of taste perception and feeding adaptations in primates.
---

## [Editor Report · Acceptance letter]

19 Sep 2023

PONE-D-23-22500R1 

Is sugar as sweet to the palate as seeds are appetizing to the belly?
Taste responsiveness to five food-associated carbohydrates in
zoo-housed white-faced sakis, *Pithecia pithecia*

Dear Dr. Laska:

I'm pleased to inform you that your manuscript has been deemed suitable for publication in PLOS ONE. Congratulations! Your manuscript is now with our production department. 

Kind regards, 

on behalf of

Dr. Nicoletta Righini 

Academic Editor

PLOS ONE